# Catalytic Technologies for Solving Environmental Problems in the Production of Fuels and Motor Transport in Kazakhstan

**Alma Massenova \*, Maxat Kalykberdiyev, Alexandr Sass, Nail Kenzin, Abzal Ussenov, Amankeldi Baiken and Kenzhegul Rakhmetova**

JSC "Institute of Fuel, Catalysis and Electrochemistry after D.V. Sokolsky", 142 D.Kunayev street, Almaty 050010, Kazakhstan; mkalykberdiev@mail.ru (M.K.); aleksandr-sass@mail.ru (A.S.); nailkenz@gmail.com (N.K.); abzalu@gmail.com (A.U.); amankeldi89@mail.ru (A.B.); rahmetova_75@mail.ru (K.R.)

\* Correspondence: almasenova@mail.ru; Tel.: +7-7772681552

**Abstract:** This research is devoted to solving an environmental problem, cleaning of the Kazakhstan air basin, through treatment of auto-transport toxic exhaust by improving the hydrocarbon composition of motor fuels and neutralizing exhaust gas toxic components. The catalytic hydrodearomatization of gasoline fractions (from the reforming stage) of the Atyrau and Pavlodar Refineries and the neutralization of exhaust gas toxic components from an internal combustion engine (ICE) were studied. Two hydrotreated gasoline fractions were tested during ICE operation. The research shows that 100% benzene conversion is observed over Rh-Pt(9:1)/$\gamma$-Al$_2$O$_3$ catalysts; that is, benzene is completely removed from both fractions, and the aromatics content decreases from 56.24–58.12% to 21.29–21.89%, within the values of the Euro-5,6 standard. Catalytic treatment of fuels reduces fuel consumption of the ICE engine by 2–3% compared to the initial gasoline fractions, the CO content in the exhaust gases decreases by 6.6–16.2%, and the hydrocarbon content decreases by 7.8–24.7%. In order to neutralize the ICE exhaust gas toxic components, the catalyst 10% Co + 0.5% Pt/Al$_2$O$_3$ was used, with which the CO conversion reaches 100% and the hydrocarbon conversion 94.2% and 91.5% for both gasoline fractions. The catalysts were characterized by electron microscopy (EM), X-ray diffraction (XRD), Brunauer–Emmett–Teller (BET), thermoprogrammed desorption (TPD) and thermoprogrammed reduction (TPR) methods. It was shown by the TPD and EM methods that at the addition of Pt to the Rh-catalyst, the formation of mixed bimetallic Rh-Pt-agglomerates occurs, and hydrogen appears in the TPD spectrum, adsorbed in the form of a new single peak uncharacteristic for the Rh-catalyst. This leads to high activity and selectivity in the hydrogenation of benzene and aromatic compounds in the gasoline fractions. The XRD and TPR results show the formation of CoAl$_2$O$_4$ spinels, on which inactive oxygen is formed for the oxidation of CO and hydrocarbons. Modification of the catalyst by Pt and Mg prevents spinel formation, thereby increasing the activity of the catalysts.

**Keywords:** rhodium-platinum catalyst; cobalt-platinum catalyst; hydrodearomatization; oxidation; gasoline fractions; exhaust gases; neutralization

## 1. Introduction

Automobile transport is the main source of air pollution globally. Millions of tons of toxic substances are emitted from exhaust gases of vehicles every year. At the present time, up to 70–80% of the pollution of the air basin of large cities is due to motor vehicles. In the city of Almaty

(Kazakhstan), the annual toxic emissions from transport represent over 150 thousand tons of carbon oxide, about 30 thousand tons of hydrocarbons, and 12 thousand tons of carbon dioxide.

An important factor in atmospheric pollution is the quality of fuels, on which depends the composition and amount of toxic exhaust gases emitted. The toxicity of types of motor gasoline and their combustion products are determined mainly by the hydrocarbon composition; namely, the content of benzenes, aromatic hydrocarbons and olefins.

Aromatic hydrocarbons in fuels include monoaromatic compounds—benzene, toluene and xylene isomers—and polyaromatic compounds—naphthalene, tetralin and other condensed aromatic compounds. Benzene is especially dangerous, as it is the most volatile and slow burning in the engine, and it is also the most chemically stable in natural conditions. The combustion of benzene forms a strong carcinogen, benzpyrene ($C_{20}H_{12}$); when 1 L of gasoline is burned, up to 81 μg of benzpyrene is formed in the exhaust gases. The higher the content of aromatic hydrocarbons in the gasoline, the higher its combustion temperature and the greater the content of nitrogen oxide in the exhaust gases [1]. More than 75% of benzene in the air comes from the exhaust gases of cars [2]. In addition, benzene enhances carbon formation in the engine and increases the soot content in the exhaust gas, which leads to a decrease in engine life [3]. The high carcinogenicity of benzene makes it necessary to limit its concentration in gasoline. Euro standards EURO-5,6 provide for a benzene content of less than 1.0%, and aromatic hydrocarbon content of less than 24%.

The largest source of benzene in gasoline is the product of the reforming process at the refinery [4,5]. Methods for changing the chemical structure of benzene and aromatics in the product and feedstock of catalytic reforming are catalytic processes [6]. The most optimal and efficient way is the catalytic hydrogenation of benzene and aromatic hydrocarbons in fuel fractions, which makes it possible to increase the environmental friendliness and improve the operational characteristics of motor fuels [2,5,7–15]. The development of catalytic technology for decreasing aromatics (hydrodearomatization) in fuel fractions of oils and fuels will improve the operational properties of domestic types of gasoline, and the environmental situation in Kazakhstan (cleaning the air basin). It should be noted that in connection with the deterioration of the quality of oil (heavy and high-sulfur oil) and the tightening of environmental requirements, the role of catalytic hydrotreating processes in oil refining processes is becoming more important.

Scientists from the USA, West Europe, Japan, Russia and China are working on solving the problem of the hydrodearomatization of refined products [16–29]. In the oil refining industry, hydrogenation processes are carried out under harsh conditions (high temperatures and hydrogen pressure) on metal oxide catalysts, where Co, Mo, Ni, Cu, W and other transition metals are used. Catalytic systems based on platinum group metals, especially Pt, Pd, Rh and Ru, are the most effective and selective catalysts for hydro-dehydrogenation reactions. In the industry, the use of both catalysts on the basis of metals of group VIII and sulphidics is common; however, the hydrogenation of benzene and aromatic hydrocarbons is now typically carried out in more stringent conditions. Recently Pt-Pd catalysts have been intensively used for the hydrotreating of petroleum products, especially for the reduction of benzene in gasoline and aromatics in diesel fuels, so they are being closely watched by researchers. The addition of Pd to $Pt/Al_2O_3$ leads to an increase of the activity and the stability in the hydrogenation of benzene. By varying the nature of the support and modifier, the catalysts reach a uniform distribution of the metals, which supports optimum acidity and stability against sulfur-containing compounds.

When gasoline is burned in the ICE of vehicles, the main toxic components are carbon monoxide, unburned light hydrocarbons, soot, sulfur and nitrogen oxides. For Kazakhstan, as for all countries, the problems of air pollution have been and remain relevant [30,31]. In Kazakhstan, about 2 million vehicles emit more than 4 million tons of harmful emissions every day. The level of air pollution of many industrial cities in Kazakhstan is more than 6–10 times higher than the existing regulatory limits. Complete catalytic oxidation of CO to $CO_2$, hydrocarbons to $CO_2$ and $H_2O$ and sulfur dioxide to trioxide is one of the most effective ways to neutralize the harmful emissions of motor transport [32–34].

Catalytic neutralizers can be divided into three main groups: Those containing noble metals; those consisting of transition metal oxides; and mixed catalysts including d-element oxides and platinum group metals. However, catalysts using noble metals remain the most common and are widely used for industrial cleaning of waste gases, despite the high cost of platinum metals.

The question of the nature and interaction of the active components of the catalysts, as well as the mechanism of the catalytic reaction [35], have been widely discussed. The results of numerous studies have shown that catalytic systems based on cobalt have catalytic activity at low temperatures, because of the unique activity for the oxidation of CO [36–42].

Investigation of the effect of alumina on the $Co_3O_4/\gamma$-$Al_2O_3$ catalyst [43–47] of various synthesis methods has shown that the catalyst obtained by impregnation methods has high activity. It was shown by the XRD method that $CoAl_2O_4$, $Co_3O_4$, Co and $Al_2O_3$ phases are present in $Co/Al_2O_3$ catalysts, depending on the Co content [48].

The catalysts obtained by the combination of Co with noble metals, in particular Pt and Co oxides, were investigated [49,50]. Noble metal promoters improved the catalytic activity and reducibility of $Co/Al_2O_3$ catalysts. It was found that the presence of platinum not only accelerates the reduction process, but also changes the sequence of the individual stages of recovery of complex spinel structures formed both during the preparation of the impregnating catalyst and during the reduction [51].

The leading manufacturers of environmental protection catalysts among non-CIS countries are UOP, Engelgard (USA), Imperial Chemical Industry (ICI, England), Haldor Topse (Denmark), Girdler (Sweden) and Rhone Poulene (France). These firms are constantly improving catalysts for cleaning exhaust gases, as the requirements for environmental protection become more stringent.

There is one way to solve the described environmental problem—vehicles must become environmentally friendly. Firstly, it is necessary to improve the quality of gasoline, and secondly it is necessary to use an effective system for neutralizing exhaust gases from toxic emissions.

The purpose of this work was to study the catalytic hydrodearomatization of two gasoline fractions from Kazakhstan refineries in order to reduce the content of benzene and aromatics, test the hydrotreated fuels in ICEs, study the composition (CO and hydrocarbons) of exhaust gases from the ICEs and purify them on a catalytic neutralizer based on metal blocks of a cellular structure.

## 2. Results and Discussion

### 2.1. Hydrogenation Reaction

The process of hydrodearomatization was studied using two gasoline fractions taken from refineries in Kazakhstan in 2020: The stable catalysate of "Atyrau Refinery" LLP (stable catalysate AR) and the stable catalysate of "Pavlodar Oil Chemistry Refinery" LLP (stable catalysate PR). Bimetallic Rh-Pt(9:1)/$\gamma$-$Al_2O_3$ catalysts were used, which showed the best results for benzene hydrogenation with 100% conversion and 100% yield of cyclohexane [52].

In gasoline fractions, benzene is hydrogenated to cyclohexane, aromatic hydrocarbons to the corresponding cycloalkanes and olefinic hydrocarbons to alkanes. The content of benzene, aromatic hydrocarbons and olefins in different gasoline fractions in comparison with the Euro-5,6 standard is shown in Table 1. The content of benzene (4.25%) and aromatics (up to 58.12%) in gasoline fractions is high because these fractions are obtained in the reforming process. These fractions are taken before the compounding process and do not contain any additives.

**Table 1.** The content of benzene and aromatic hydrocarbons in gasoline fractions of AR and PR in comparison with Euro-5,6.

| Name of the Fraction | Benzene Content, Mas. % | Aromatics Content, Mas. % | Olefin Content, Mas. % |
|---|---|---|---|
| Euro-5,6 | <1 | 24 | 5 |
| Stable Catalysate AR | 4.25 | 56.24 | 0.58 |
| Stable Catalysate PR | 1.31 | 58.12 | 11.02 |

The effect of the content of the active catalytic phase on the composition of the components of two gasoline fractions during hydrodearomatization over catalysts 0.1% Rh-Pt/γ-Al$_2$O$_3$, 0.5% Rh-Pt/γ-Al$_2$O$_3$ and 1.0% Rh-Pt/γ-Al$_2$O$_3$ at 100 °C and 4.0 MPa was studied (Figure 1). In the process of hydrodearomatization of stable catalysate AR, with an increase of the active phase content, the conversion of aromatic hydrocarbons increased, and their content decreased from 56.24% to 32.82% for the 0.1% Rh-Pt/γ-Al$_2$O$_3$ catalyst, to 27.92% for the 0.5% Rh-Pt/γ-Al$_2$O$_3$ catalyst and to 24.01% for the 1.0% Rh-Pt/γ-Al$_2$O$_3$ catalyst.

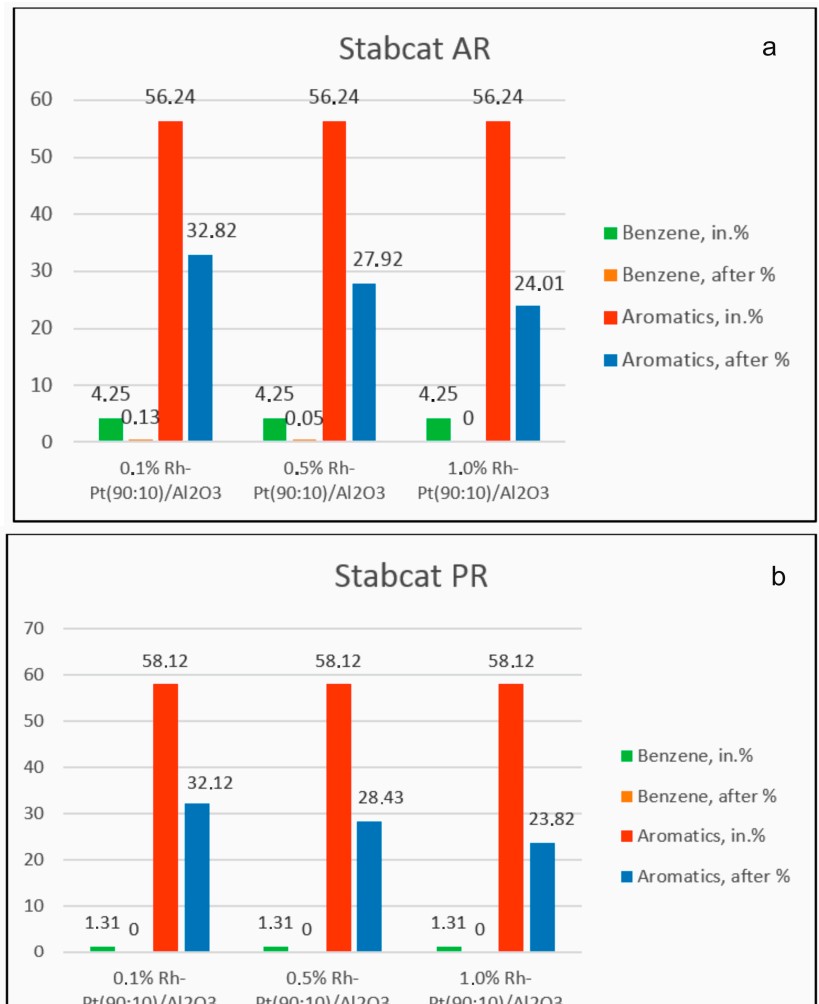

**Figure 1.** Hydrogenation of fractions of stable catalysate AR (**a**) and stable catalysate PR (**b**) over 0.1% Rh-Pt/γ-Al$_2$O$_3$, 0.5% Rh-Pt/γ-Al$_2$O$_3$ and 1.0% Rh-Pt/γ-Al$_2$O$_3$ catalysts at 100 °C and 4.0 MPa.

For the stable catalysate PR, the content of aromatics also decreased from 58.12% to 32.12% in the case of the 0.1% Rh-Pt/γ-Al$_2$O$_3$ catalyst, to 28.43% for 0.5% Rh-Pt/γ-Al$_2$O$_3$ and to 23.82% for 1.0% Rh-Pt/γ-Al$_2$O$_3$. Over all three catalysts for both types of gasoline fractions, benzene was completely removed from the fraction.

The influence of technological parameters of the hydrodearomatization process of gasoline fractions (pressure 1–5 MPa, temperature 25–200 °C) on the content of benzene and aromatic hydrocarbons was also studied. At temperatures of 25–200 °C and hydrogen pressures of 2–5 MPa, benzene was completely removed from the two fractions, and the amount of aromatic hydrocarbons was reduced by 1.5–2.6 times (Table 2). During gasoline hydrogenation of stable catalysate AR, the content of aromatic hydrocarbons at 25–200 °C decreased from 56.24% to 21.29%, and with increasing pressure from 1.0 to 5.0 MPa it decreased from 56.24% to 22.87%. During hydrogenation of stable catalysate PR in the

temperature range of 25–200 °C the aromatics content decreased from 58.12% to 21.98%, and when the pressure changed from 1.0 to 5.0 MPa it decreased from 58.12% to 21.89%.

**Table 2.** Hydrodearomatization of fractions of stable catalysate AR and stable catalysate PR over the 0.5% Rh-Pt/$\gamma$-Al$_2$O$_3$ catalyst.

| Conditions | | Benzene, Mas. % | | Aromatics, % Mas. | |
|---|---|---|---|---|---|
| | | Initial, % | After Exp., % | Initial, % | After Exp., % |
| **Stable Catalysate AR** | | | | | |
| P, MPa at 50 °C | 1.0 | | 3.75 | | 39.89 |
| | 2.0 | | 2.02 | | 29.67 |
| | 3.0 | 4.25 | 0 | 56.24 | 24.01 |
| | 4.0 | | 0 | | 23.45 |
| | 5.0 | | 0 | | 22.87 |
| T, °C at 4 MPa | 25 | | 3.34 | | 43.57 |
| | 50 | | 1.89 | | 32.25 |
| | 100 | 4.25 | 0 | 56.24 | 24.01 |
| | 150 | | 0 | | 22.98 |
| | 200 | | 0 | | 21.29 |
| **Stable Catalysate PR** | | | | | |
| P, MPa at 50 °C | 1.0 | | 1.19 | | 35.24 |
| | 2.0 | | 1.02 | | 28.87 |
| | 3.0 | 1.31 | 0 | 58.12 | 23.82 |
| | 4.0 | | 0 | | 22.15 |
| | 5.0 | | 0 | | 21.89 |
| T, °C at 4 MPa | 25 | | 1.24 | | 39.56 |
| | 50 | | 0.57 | | 29.57 |
| | 100 | 1.31 | 0 | 58.12 | 23.82 |
| | 150 | | 0 | | 22.87 |
| | 200 | | 0 | | 21.98 |

Data on the group composition of organic substances in two types of gasoline, in the initial fractions and after hydrogenation over Rh-Pt(9:1)/$\gamma$-Al$_2$O$_3$ at 3 MPa and 100 °C, are shown in Figure 2. For the stable catalysate AR (Figure 2a) the benzene content in the initial state was 4.25%, after the reaction benzene was absent, representing 100% benzene conversion. The number of aromatics decreased from 56.24% to 24.01%. It should be noted that the number of olefins decreased from 0.58% to 0%, which is very favorable for gasolines, since the presence of olefins leads to instability (the oligomerization and polymerization reaction proceeds). The amount of paraffins practically did not change, going from 12.22 to 12.02%. The content of isoparaffins increased from 24.03% to 36.76%. Apparently, the isomerization of paraffins to isoparaffins occurred. The content of naphthenes increased sharply from 2.68% to 27.21%.

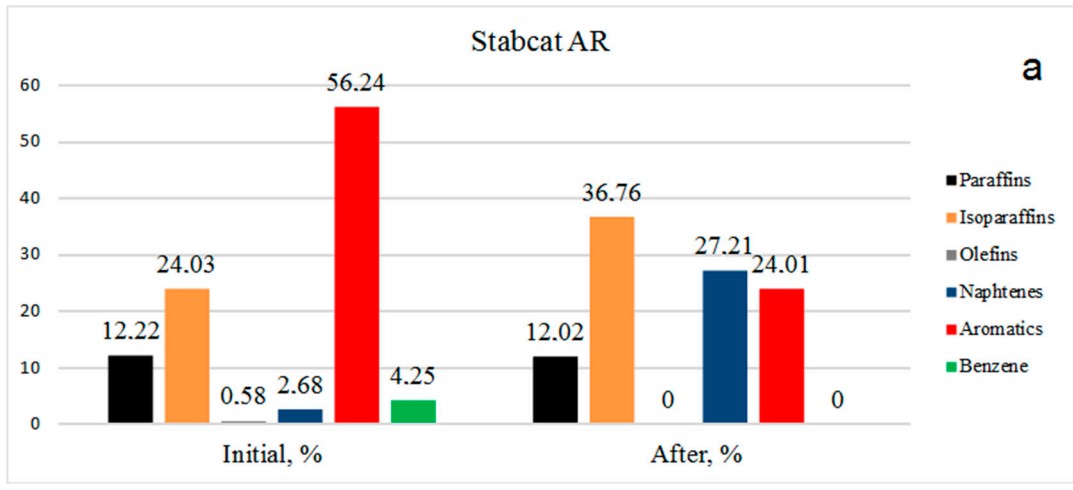

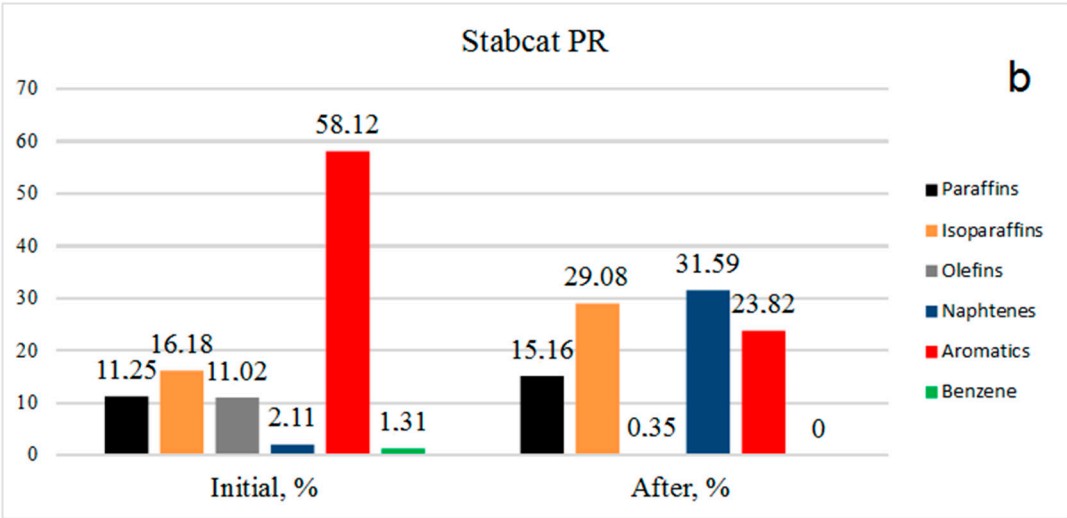

**Figure 2.** Group composition of gasoline fractions over Rh-Pt/$\gamma$-Al$_2$O$_3$: (**a**) Stable catalysate AR, (**b**) stable catalysate PR.

The picture is similar for stable catalysate PR (Figure 2b): Benzene was absent, aromatics decreased from 58.12% to 23.82%, olefins decreased from 11.02% to 0.35%, isoparaffins increased from 16.18% to 29.08% and the amount of paraffins increased due to the hydrogenation of olefins from 11.25% to 15.16%. It should be noted that an increase in the content of paraffins of the isostructure indicates hydrogenation reaction implementation, but also hydroisomerization.

Data on the octane number and the density of the gasoline fractions before and after the catalytic treatment are given in Table 3. According to the research method (RM), the octane number after the treatment with stable catalysate AR was unchanged, and was equal to 94.3–94.2 units. The octane number, according to the motor method (MM), increased from 83.2 to 83.3; that is, in this case also, the octane number was unchanged. For stable catalysate PR, the octane number decreased from 90 to 89.9 (RM). This indicates that the processing of gasoline practically does not affect the octane number. The density slightly increased after treatment, which is understandable from the point of view of changing the hydrocarbon composition to a heavier region; naphthenes have a higher density compared to aromatic hydrocarbons.

**Table 3.** Characteristics of gasoline fractions of stable catalysate AR and stable catalysate PR before and after catalytic treatment.

| Gasoline | Octane Number RM | Octane Number MM | Density, g/cm$^3$ |
|---|---|---|---|
| | **Stable Catalysate AR** | | |
| Initial | 94.3 | 83.2 | 0.771 |
| After Experiment | 94.2 | 83.3 | 0.780 |
| | **Stable Catalysate PR** | | |
| Initial | 90.0 | - | 0.793 |
| After experiment | 89.9 | - | 0.798 |

Data on the characteristics of catalysts obtained by BET method and porosimetry are given in Table 4, which shows the surface area of the catalysts Pt/γ-Al$_2$O$_3$, Rh/γ-Al$_2$O$_3$ and Rh-Pt(9:1)/γ-Al$_2$O$_3$.

**Table 4.** Characteristics of the prepared supported catalysts.

| Catalyst | Me Wt. % | $S_{total}$ m$^2$/g | d, Å |
|---|---|---|---|
| Pt/γ-Al$_2$O$_3$ | 0.5 | 175 | 12–25 |
| Rh/γ-Al$_2$O$_3$ | 0.5 | 144 | 4–14 |
| Rh-Pt(9:1)/γ-Al$_2$O$_3$ | ΣMe = 0.1 | 131 | 6–15 |
| Rh-Pt(9:1)/γ-Al$_2$O$_3$ | ΣMe = 0.5 | 138 | 8–18 |
| Rh-Pt(9:1)/γ-Al$_2$O$_3$ | ΣMe = 1.0 | 129 | 7–18 |

For hydrogenation, it is very important to know the state of adsorbed hydrogen on the catalyst surface. The curves of thermal desorption of hydrogen with the 0.5% Rh/Al$_2$O$_3$, Rh-Pt catalysts used in this work are shown in Figure 3. Two regions of desorption were observed for 0.5% Rh/Al$_2$O$_3$, with $T_{max}$ 168 and 401 °C (curve 1) corresponding to the low- and tightly bound nature of the hydrogen forms with the surface. The addition of Pt into the Rh-catalyst in the ratio of 9:1 led to a significant change in the TPD spectrum (curve 3). A new region of hydrogen desorption appeared in the form of a single peak with $T_{max}$ 278 °C, located between the high and low temperature regions, and a small shoulder in the region of 400–500 °C also appeared. The appearance of a new peak may be due to the adsorption of hydrogen on mixed bimetallic Rh-Pt agglomerates, as will be shown below in the EM images. By increasing the amount of Pt to the ratio of 5:5 (curve 2), the character of the TPD spectrum for Rh-Pt catalysts does not change; there is a rather intense peak with $T_{max}$ 292 °C and a weaker low temperature peak with $T_{max}$ 212 °C. It should also be noted that the total intensity of the peaks antibatically decreases with increasing Pt content. Apparently, hydrogen is sorbed mainly on the Rh surface and decreases symbatically with decreasing Rh content. The high homogeneity of the absorbed hydrogen (the main part is desorbed in one region with $T_{max}$ 278–292 °C) leads to a high activity of 0.5% Rh-Pt/A1$_2$O$_3$ catalysts in the test process.

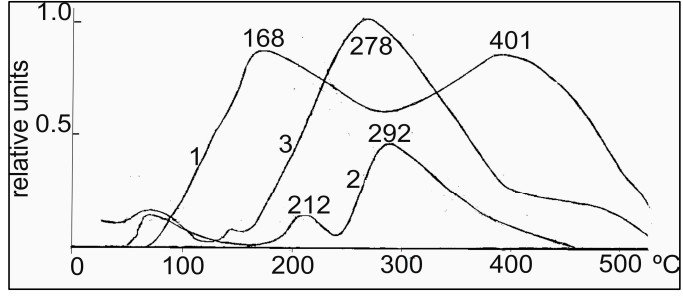

**Figure 3.** Thermal desorption of hydrogen from various catalysts in the mode of linear temperature increase from 0 to 750 °C: 1. 0.5% Rh/γ-Al$_2$O$_3$; 2. 0.5% Rh-Pt(5:5)/γ-Al$_2$O$_3$; 3. 0.5% Rh-Pt(9:1)/γ-Al$_2$O$_3$.

The surface of the catalysts was studied using SEM and TEM methods. Figure 4 shows the SEM image of the Rh-Pt(9:1)/Al$_2$O$_3$ catalyst. It shows a sufficiently uniform surface of the support, on which agglomerates of active metals are visible. TEM images of the catalyst with different contents of supported metals are shown in Figure 4b–d. The catalyst Rh-Pt(9:1)/Al$_2$O$_3$ is represented by finely dispersed particles, 2–25 nm in size, and in addition there is a small amount of denser and larger particles of 5 nm size on the surface, the microdiffraction patterns of which are represented by diffuse rings corresponding to the metals Pt° and Rh° (Figure 4b–d). With an increase in the concentration of supported metals in the indicated range, as can be seen from this figure, the particle sizes do not change, but the filling density of the support surface increases. Metal particles are distributed on the surface of the alumina and mixed bimetallic agglomerates of Rh-Pt, together with Rh particles, are presented. Apparently, the high activity of the catalyst is due to the uniform distribution of nano-sized metal particles in the zero valence state on the surface of the alumina.

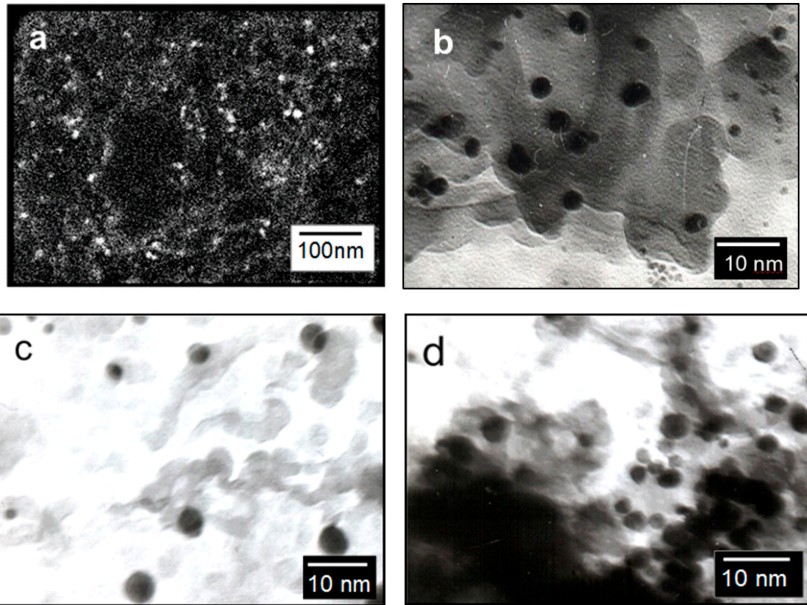

**Figure 4.** Electron microscopy (EM) images of Rh-Pt(9:1)/γ-Al$_2$O$_3$. (**a**) SEM. (**b–d**) TEM (an increase of 160,000) at Rh-Pt (9:1) concentration. (**a,b**) 0.5%. (**c**) 0.1%. (**d**) 1.0%.

The diffractogram of the Rh-Pt(9:1)/γ-Al$_2$O$_3$ catalyst is shown in Figure 5. The diffractogram shows reflections of 4.69, 2.77, 2.41, 2.29, 1.98, 1.51, 1.39 Å (JCPDS 10-425) related to γ-Al$_2$O$_3$. The absence of reflections of Pt 2.26 Å (JCPDS 4-802) and Rh 2.20 Å (JCPDS 5-68) indicates significant X-ray amorphism, and dispersion of Rh and Pt on the support surface.

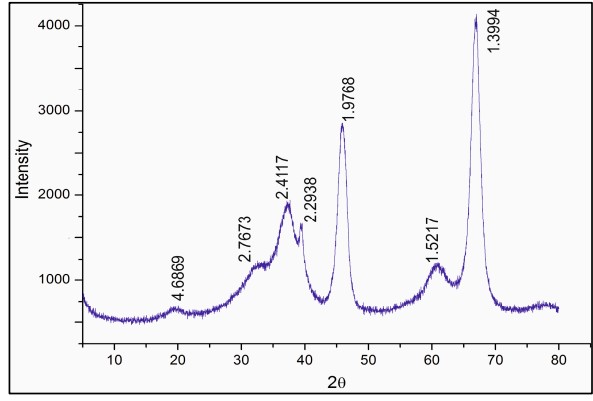

**Figure 5.** Diffractogram of the Rh-Pt(9:1)/γ-Al$_2$O$_3$ catalyst.

## 2.2. Testing Gasoline Fractions in an ICE

Four types of fuel—initial gasoline fractions of stable catalysate AR and stable catalysate PR before and after catalytic hydrodearomatization—were tested with a KIPOR KG160 ICE. The main purpose of this test was to measure the toxic components of the ICE exhaust, specifically carbon monoxide and hydrocarbons, using an Optima 7 gas analyzer.

The engine gas tank was filled with 2.5 kg of each type of fuel, and by the time of operation at an average speed of 2500 rpm, the engine operation time was measured until it was completely stopped or the fuel was used up. The start of the ICE operation was counted from a cold engine. The operating time of the ICE with each of the four types of gasoline is shown in Table 5. The operating time using treated fuels increased by 3% for stable catalysate AR and by 2% for stable catalysate PR. Thus, cleaning fuels of benzene and aromatic hydrocarbons leads to a decrease in fuel consumption.

**Table 5.** Time of internal combustion engine (ICE) operation until complete stop.

| Type of Fuel | | Time, Min. |
|---|---|---|
| Stable Catalysate AR | Initial | 92 |
| | After Experiment | 95 |
| Stable Catalysate PR | Initial | 90 |
| | After Experiment | 92 |

Ten minutes after the start of engine operation at 2500 rpm, the components of the exhaust gases of the ICE were measured using a gas analyzer. The analysis results are shown in Table 6.

**Table 6.** The amount of toxic substances contained in the exhaust gases of the ICE.

| Component | Stable Catalysate AR | | Stable Catalysate PR | |
|---|---|---|---|---|
| | Initial | After Exp. | Initial | After Exp. |
| CO, ppm | 8162 | 7621 | 9473 | 7938 |
| $CO_2$, vol. % | 11.6 | 11.0 | 10.4 | 11.2 |
| $O_2$, vol. % | 3.2 | 2.9 | 2.5 | 2.3 |
| $CH_x$, ppm | 1647 | 1518 | 2376 | 1789 |
| $SO_2$, ppm | 0 | 0 | 0 | 0 |
| NO, ppm | 151 | 146 | 180 | 191 |
| $NO_2$, ppm | 4 | 2 | 1 | 5 |

The gas analyzer data (Table 6) showed that for the gasoline fraction of stable catalysate AR after catalytic hydrodearomatization, the amount of carbon monoxide decreased from 8162 ppm to 7621 ppm (by 6.6%). The content of hydrocarbons ($CH_x$) also decreased from 1647 ppm to 1518 ppm (by 7.8%). The amount of nitrogen oxides slightly decreased from 151 ppm to 146 ppm. In the case of stable catalysate PR, the CO content decreased from 9473 ppm to 7938 ppm (by 16.2%). The amount of hydrocarbons decreased from 2376 ppm to 1789 ppm (by 24.7%). It should be noted that sulfur dioxide is not contained in the initial fuels, apparently due to the complete removal of sulfur compounds during processing at the refinery.

Thus, hydrotreated gasoline fractions from Kazakhstan refineries, during combustion, form less toxic components in the exhaust gases than the initial fuels.

## 2.3. Oxidation Reaction (Neutralization)

The next stage of the study was the neutralization of toxic components of the exhaust gases of ICEs from the combustion of hydrotreated gasoline fractions on a catalytic neutralizer. A block of platinum-cobalt catalyst was tested, as it showed the highest activity in early studies [53] in the oxidation of carbon monoxide. It was shown that CO can be completely removed from the exhaust gas.

Catalyst 10%Co + 0.5%Pt/Al$_2$O$_3$ was placed in a metal case in front of the exhaust collector of the ICE to test the oxidation of carbon monoxide and hydrocarbons (CH$_x$) in the composition of ICE exhaust gases, the results of which are presented in Table 7.

**Table 7.** Oxidation of CO and CH$_x$ during neutralization of the exhaust gases of an ICE over a 10% Co + 0.5% Pt/Al$_2$O$_3$ catalyst.

| Temperature, °C | Stable Catalysate AR | | Stable Catalysate PR | |
|---|---|---|---|---|
| | Ppm | Conversion ($\alpha$), % | Ppm | Conversion ($\alpha$), % |
| | | CO | | |
| 50 | 7821 | 0.0 | 7938 | 0.0 |
| 100 | 7054 | 9.8 | 7247 | 8.7 |
| 150 | 3026 | 61.3 | 3468 | 56.3 |
| 200 | 94 | 98.8 | 341 | 95.7 |
| 250 | 0 | 100 | 0 | 100 |
| | | CH$_x$ | | |
| 50 | 1518 | 0 | 1789 | 0 |
| 100 | 1433 | 5.6 | 1701 | 4.9 |
| 150 | 776 | 49.9 | 953 | 46.7 |
| 200 | 322 | 79.8 | 432 | 75.8 |
| 250 | 88 | 94.2 | 152 | 91.5 |

The studies were carried out in a temperature range of 50–250 °C and a space velocity of 10,000 h$^{-1}$. The process does not proceed at all at 50 °C and the conversion rate is 0%, while the CO and CH$_x$ contents remain unchanged. With an increase in temperature, the degree of conversion of carbon monoxide for stable catalysate AR increased from 9.8% at 100 °C, and at 250 °C reached 100%, and from 8.7% to 100% in the case of stable catalysate PR. This means the CO was removed completely from both types of gasoline.

At 50 °C the CH$_x$ conversion rate to CO$_2$ and H$_2$O is 0%, and the CH$_x$ content is the same as in the initial product. For stable catalysate AR, with an increase in temperature the degree of conversion of CH$_x$ increases from 100 °C (5.6%), and at 250 °C reaches 94.2%, at which point the content of CH$_x$ is 88 ppm. In the case of stable catalysate PR, at 100 °C conversion of CH$_x$ is 4.9% and at 250 °C it is 91.5%, and the amount of CH$_x$ is 152 ppm.

Thus, according to the test results, 100% CO conversion and 91.5% and 94.2% CH$_x$ conversion were obtained in the process of neutralizing exhaust gases from the ICE at various temperatures, for two hydrotreated types of gasoline.

To study the activity of the platinum-cobalt catalyst, TPR and XRD tests of the samples were carried out. This provides information about the forms of oxygen that are active during the oxidation process.

2.3.1. Study of Catalysts by the XRD Method

Diffractograms of samples of the 10% Co/Al$_2$O$_3$ catalysts calcined at 550, 850 and 1100 °C are shown in Figure 6. There are reflections at 550 °C which are characteristic for Co$_3$O$_4$ cobalt-cobalt spinels (JCPDS 73-1701). It should be noted that cobalt aluminate, CoAl$_2$O$_4$ (JCPDS 3-896), has a close set of reflections. These compounds differ only in the reflection at 4.6670 Å, which is characteristic of the two of them only for Co$_3$O$_4$: 4.67, 2.86, 2.44, 2.02, 1.55 and 1.43 Å. Reflections of Al$_2$O$_3$ are absent. Heating the sample at 850 °C in air leads to a decrease of reflection 4.67 Å and the preservation of the set of reflections from Co$_3$O$_4$ and CoAl$_2$O$_4$. Reflections of $\gamma$-Al$_2$O$_3$ (JCPDS 10-425) are not explicitly expressed due to the rather high dispersion, and $\alpha$-Al$_2$O$_3$ (JCPDS 10-173) is absent. At 1100 °C, the reflection 4.7 Å from Co$_3$O$_4$ is at the noise level and, apparently, this phase is absent. There are narrow intense reflections from CoAl$_2$O$_4$ on the roentgenogram, as well as narrow reflections from $\alpha$-Al$_2$O$_3$: 3.50, 2.56, 2.38, 2.09, 1.74, 1.60, 1.54, 1.40, 1.37 and 1.24 Å. This indicates that if Co$_3$O$_4$ is

present in the sample at 550 °C, then at 850 °C the intensity of the maximum decreases significantly. At 1100 °C, almost all cobalt oxide is converted into cobalt–aluminum spinels.

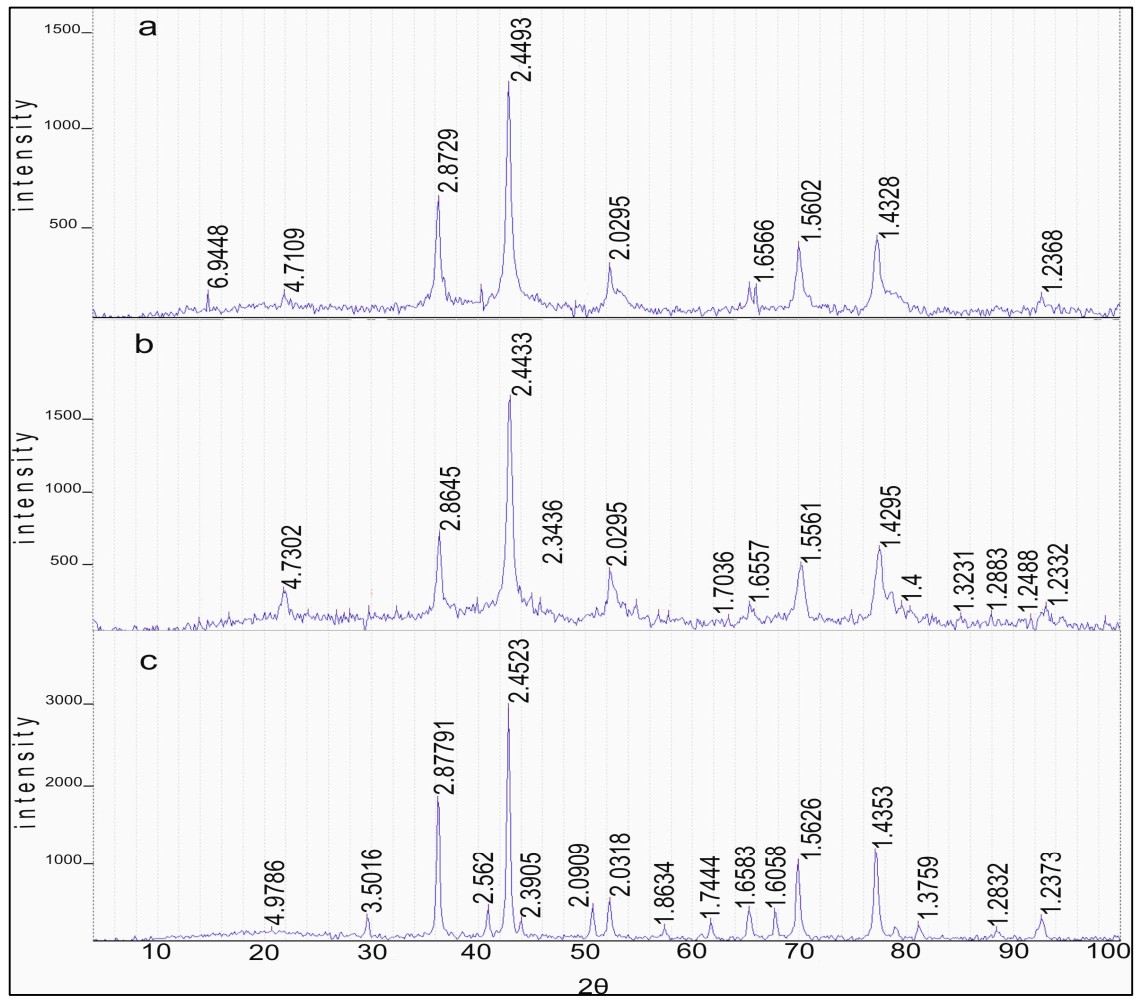

**Figure 6.** XRD patterns of 10% Co/Al$_2$O$_3$ calcined in air at (**a**) 550, (**b**) 850, (**c**) 1100 °C.

In Figure 7, the same samples are shown after treatment in a reducing medium with hydrogen in the TPR mode, after removal of all forms of oxygen from the surface of the catalyst. The samples were cooled to room temperature in the stream of hydrogen, and then their diffractograms were taken. Since during the TPR experiment the samples were heated from room temperature to 1100 °C, the support undergoes some change due to the partial transition from γ-Al$_2$O$_3$ to the α-Al$_2$O$_3$ form. This is invisible at 550 °C, and at 850 °C there are partial reflections from α-Al$_2$O$_3$. At the same time, all the samples are characterized by the presence of reflections of metallic cobalt: 2.05, 1.78 and 1.25 Å (JCPDS 15-806). In the diffractogram of the sample calcined at 1100 °C, the reflections from spinels, either Co$_3$O$_4$ or CoAl$_2$O$_4$: 2.84, 2.45, 1.55 and 1.41 Å, are probably preserved in small amounts at the noise level. Reflections from α-Al$_2$O$_3$ and metallic cobalt are clearly observed. In the special experiment, we took a sample, oxidized it in the air at 1100 °C, then reduced it in the TPR mode and re-oxidized it in air at 600 °C within 1 h. As can be seen from Figure 7, 1 on the background of reflections from α-Al$_2$O$_3$: 3.49, 2.56, 2.38, 2.09, 1.74, 1.60, 1.56, 1.41, 1.38 and 1.24 Å (JCPDS 10-173), clear intense maxima from Co$_3$O$_4$ spinels are observed: 4.69, 2.87, 2.44, 2.02, 1.65, 1.56 and 1.43 Å (JCPDS 73-1701). The presence of metallic cobalt was not detected.

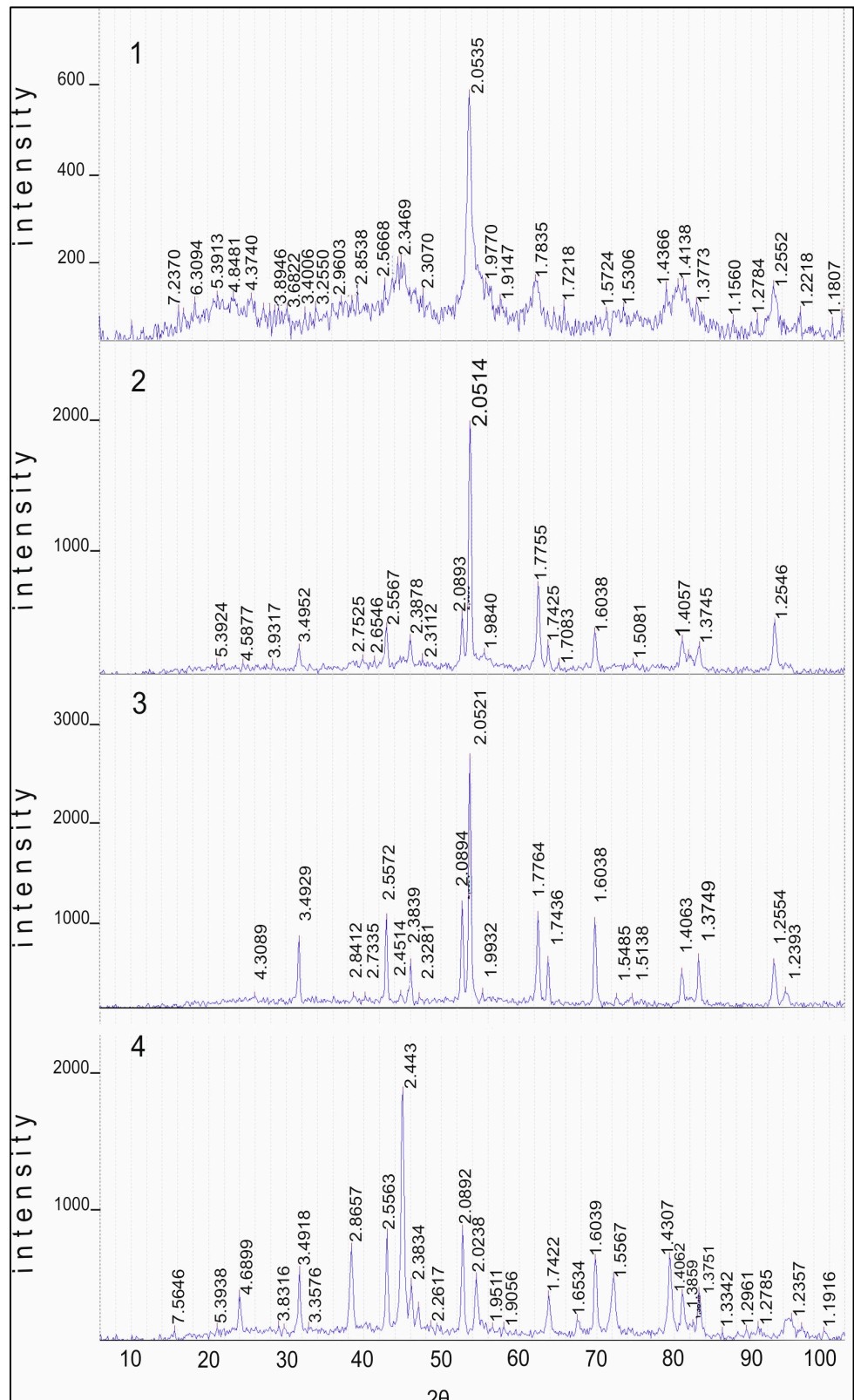

**Figure 7.** XRD patterns of 10% Co/Al$_2$O$_3$ calcined in air at (**1**) 550, (**2**) 850, (**3**) 1100 °C and then reduced in the thermoprogrammed reduction (TPR) mode up to 1200 °C. Sample (**4**) was obtained by oxidation in air at 600 °C, using the sample after TPR.

When $Co(NO_3)_2 \cdot 6H_2O$ and $Mg(NO_3)_2 \cdot nH_2O$ are co-deposited into the support and then calcined at 1100 °C for 1 h in the air, two sets of reflections are observed from the MgO phase and CoO/MgO solid solution, 2.43, 2.11, 1.48, 1.27 and 1.21 Å (JCPDS 4-829 and 2-1201) and $Co_3O_4$ 4.66, 2.86, 2.43, 2.02, 1.58 and 1.43 Å (JCPDS 73-1701) (Figure 8). A close set of reflections also have spinels: $MgCo_2O_4$ (JCPDS 2-1073) and $CoAl_2O_4$ (JCPDS 3-896). However, $Co_3O_4$, unlike these two spinels, has a reflection at 4.66 Å with an intensity of about 18%, which makes it possible to uniquely attribute the set of these reflections to the existence of the $Co_3O_4$ phase. However, we cannot exclude the existence, especially in the presence of MgO, of spinels of $MgCo_2O_4$ and $CoAl_2O_4$. It should be noted that there are no $Al_2O_3$ reflections of both the γ- and α-forms on the roentgenogram and there is no reflection from $MgAl_2O_4$, which can be formed from MgO and $Al_2O_3$ under more stringent conditions (pressure 280 Kbar and temperature 1100 °C (JCPDS 33-853).

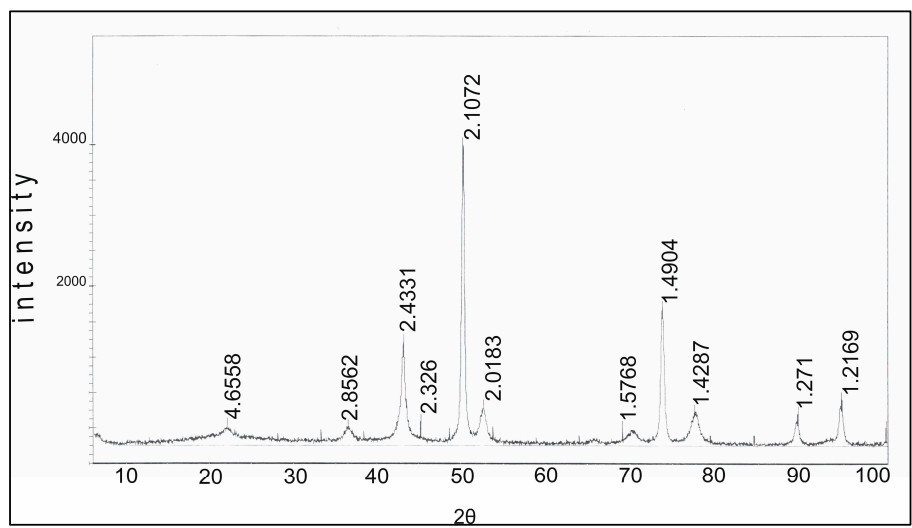

**Figure 8.** XRD patterns of 15% Co + 15% Mg/$Al_2O_3$ calcined in air at 1100 °C.

### 2.3.2. Catalyst TPR Study

Figure 9 shows TPR spectra of the Co/$Al_2O_3$ catalysts obtained by impregnating γ-$Al_2O_3$ with cobalt nitrate, with a metal content of 10% by weight in terms of the metal Co. Curve 1 corresponds to the spectrum of TPR of the catalyst, dried to the air-dry state of $Co(NO_3)_2$ on $Al_2O_3$. Conditionally, the spectrum can be divided into three regions: I up to 550 °C, II 550–900 °C and III above 900 °C.

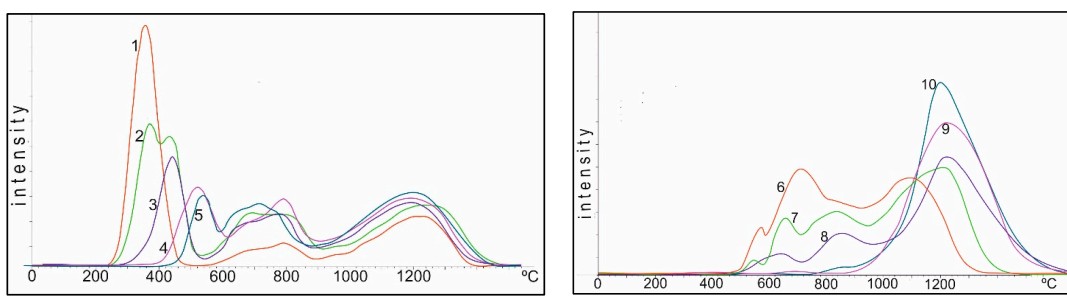

**Figure 9.** TPR data for 10% Co/$Al_2O_3$ calcined in air at 25–1100 °C; 1. 25, 2. 300, 3. 400, 4. 550, 5. 650, 6. 750, 7. 800, 8. 850, 9. 950, 10. 1100 °C. The intensity of the peaks of samples 6–10 is doubled.

In the first (I) region, when the catalyst is reduced by hydrogen, the most active oxygen of the cobalt oxide is reduced during decomposition with a linear increase in temperature. The oxygen of these oxides is most active in oxidation reactions of the toxic components of exhaust gases: CO and $C_xH_y$. As the heating temperature of the sample in air increases, the oxygen content in the second

region drops to 0 at a heating temperature of 800–850 °C, and the temperature of the maximum oxygen reduction in this region shifts from 350 to 550 °C.

In the second (II) region, the oxygen peaks are more diffuse; their total amount (area of peaks) changes little to 750 °C, then begins to drop and greatly decreases during heating between 950–1100 °C.

In the third (III) region, the amount of oxygen in the temperature range of 750–800 °C varies insignificantly, and during heating between 950–1100 °C practically reaches the maximum. The peak of oxygen in this region corresponds to the data of the XRD analysis spinel of cobalt aluminate, $CoAl_2O_4$.

Figure 10 shows the TPR curves of the 10%Co + 0.5%Pt/$Al_2O_3$ catalysts obtained by co-impregnation and calcined at temperatures of 25–1100 °C. This image can also be divided into three areas: I up to 450 °C, II 450–800 °C and III above 800 °C. This effect of shifting the temperature intervals is apparently connected to the presence of platinum, which can additionally activate hydrogen in the gas mixture of TPR. Perhaps the process of spillover of hydrogen, activated by platinum, occurs. The cardinal difference between Co-Pt-Al and Co-Al catalysts is the initial insignificant amount of oxygen in the III region. Co-Al catalysts, even at deposition at 25 °C and subsequent TPR, show a partial transition of cobalt oxides to spinel $CoAl_2O_4$. In the presence of platinum, this process is strongly suppressed up to 650 °C. This may be due to the fact that platinum promotes the conservation of cobalt in the form of catalytically active oxides, rather than formation of low-activity aluminum–cobalt spinels.

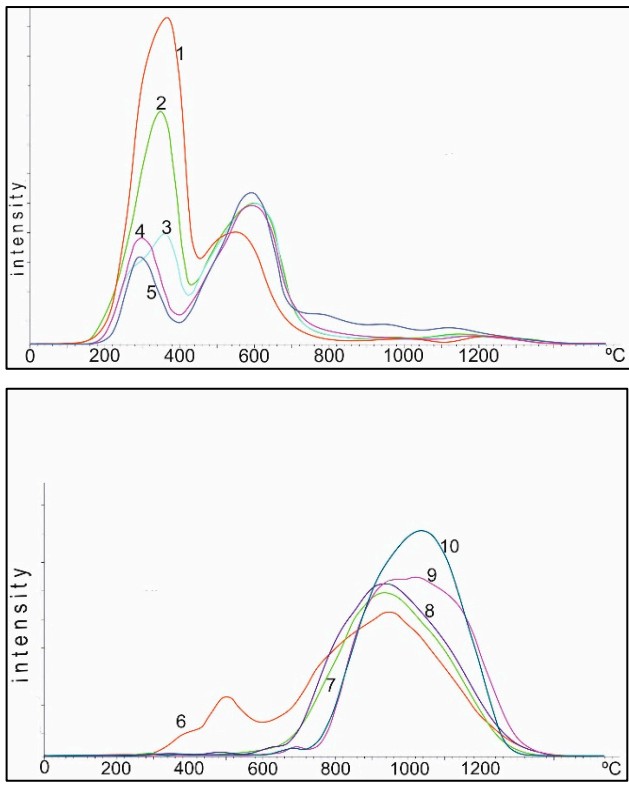

**Figure 10.** TPR data for 10% Co + 0.5% Pt/$Al_2O_3$ calcined in air at 25–1100 °C. 1. 25, 2. 300, 3. 400, 4. 550, 5. 650, 6. 750, 7. 800, 8. 850, 9. 950, 10. 1100 °C. The intensity of the peaks of samples 6–10 is doubled.

The oxygen in the I region of the Co-Pt catalysts also tends to decrease as the temperature increases, and becomes more homogeneous, but at 400–650 °C (oxidation of the catalyst in air), the temperature of reduction maximum shifts to the low-temperature region from 350 to 280–300 °C. The amount of oxygen in the II region to 650 °C (heating of the catalyst with the air) varies insignificantly, but from 750 °C it decreases sharply and at 800 °C practically disappears. Up to 650 °C, platinum prevents the formation of cobalt aluminate, and starting from 750 °C, on the contrary, it accelerates this process in comparison with aluminum–cobalt catalysts. If on Co-Al catalysts calcined in the air up to 850 °C, oxygen in the

II region is stably observed, then on Co-Pt catalysts at 800 °C this process is practically completed. Platinum on the one hand hinders, and on the other hand accelerates, the formation of spinel $CoAl_2O_4$, depending on the temperature interval. This is apparently due to the fact that while the Pt-Co complex is preserved up to 650–750 °C, the formation of $CoAl_2O_4$ is not observed, and, as the temperature increases, the cobalt aluminate formation accelerates as the complex is destroyed.

To preserve as much free cobalt oxide as possible on the surface of the catalyst, samples with a higher cobalt content were prepared. At 10% Co the cobalt–aluminum ratio is below the stoichiometric ratio to form the $CoAl_2O_4$ compound. At 30%, this ratio roughly corresponds to stoichiometry. At 50% Co it is above stoichiometry. As can be seen in Figure 11, on such samples calcined at 950 °C, with an increase in the Co concentration, the TPR peak is noticeably broadened at 30% relative to 10% Co, and an even larger broadening with a shift to the II region is observed for 50%. At the same time, oxygen of oxides of cobalt in the I and II regions is stable on the 50% sample.

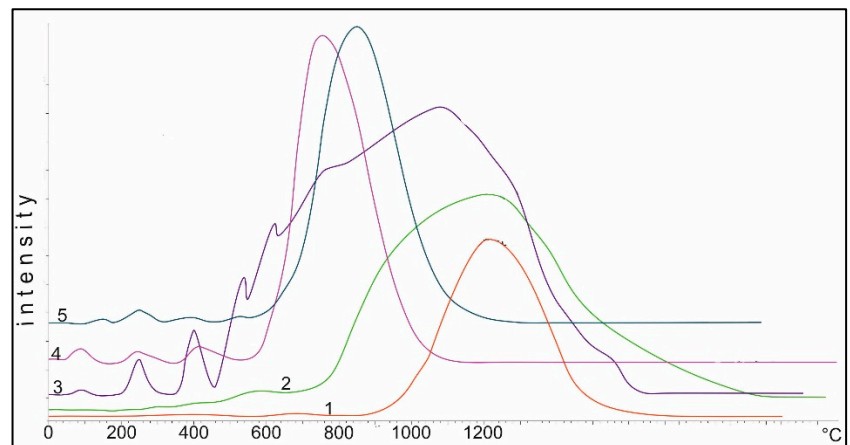

**Figure 11.** TPR data for Co/$Al_2O_3$ calcined in air at 25–1100 °C with different contents of Co (1. 10, 2. 30, 3. 50%) and with additive of 15% Mg at 950 and 1100 °C (4–5).

In order to prevent the formation of low-active Co-Al-spinels, we similarly prepared Co-Mg catalysts when the support was co-impregnated with cobalt and magnesium nitrates calcined at 950 and 1100 °C. As can be seen from Figure 11, even under such severe heating conditions, oxygen is observed in both the I and II regions, and the formation of oxygen in the III region is insignificant.

This corresponds to XRD data (Figure 8), where the main fraction of cobalt is in the form of $Co_3O_4$ (II region of TPR), partially free cobalt oxides (I region) appear, and the formation of cobalt aluminate is insignificant.

According to the TPR data, the oxygen of spinels of $CoAl_2O_4$ is less active in the reaction with hydrogen, which at temperatures above 800 °C is reduced according to XRD data on metallic cobalt. To prevent the formation of this spinel and the appearance of free cobalt oxides, even at high temperatures above 1000 °C, the support was filled with more active non-transition metals, of which the most suitable turned out to be Mg, which has a size and cation charge close to that of cobalt. Other more active metals, for example calcium with alumina, give aluminates, which in the presence of water vapor can gradually decompose. Therefore, magnesium oxide turned out to be the most convenient cobalt substitute in the spinel structure. This was confirmed by the data obtained by the TPR and XRD methods. It is also possible to prevent the formation of low-active cobalt aluminate when the cobalt content is increased to 50% (above the stoichiometry in $CoAl_2O_4$). At a lower content, the process of cobalt absorption by the support is very active (Figure 11). The study of the addition of platinum and the formation of cobalt aluminate showed a twofold effect: At temperatures up to 650 °C, platinum promotes the retention of active cobalt oxides and prevents the formation of cobalt aluminate, and after the possible decomposition of the platinum complex with cobalt oxide with increasing heating temperature, on the contrary, it catalyzes the acceleration of the formation of spinel

$CoAl_2O_4$. This effect of platinum has been confirmed in other studies [49–51] on the effect of platinum on the recovery of complex spinel structures. Thus, the addition of Pt and Mg helps to prevent the formation of the low-active spinel $CoAl_2O_4$ and preserves the more reactive $Co_3O_4$ phase.

## 3. Experimental Methods

### 3.1. Materials

The materials used were $H_2PtCl_6 \cdot 6H_2O$ (TU 2612-034-00205067-2003, LLC "Aurat", Moscow, Russia), $RhCl_3 \cdot 3H_2O$ (MRTU 609 1833-64, LLC "Aurat", Moscow, Russia) and $Co(NO_3)_2 \cdot 6H_2O$ (GOST 4528-78, PJSC MMC "Norilski nikel", Norilsk, Russia). Pseudoboehmite was used to produce $\gamma$-$Al_2O_3$ (LLC "UfaChim", Ufa, Russia). We used the stable catalysate of "Atyrau Refinery" LLP and stable catalysate of "Pavlodar Oil Chemistry Refinery" LLP.

### 3.2. Catalyst Preparation

#### 3.2.1. Catalysts for Hydrogenation

The $\gamma$-$Al_2O_3$ support for two types of catalysts (for hydrogenation and neutralization) was prepared from pseudoboehmite by calcination at 650 °C for 2 h. The specific surface area of the alumina was 170 $m^2$/g.

In this work, catalysts based on the platinum group metals Pt and Rh were used. For the preparation of catalysts, we used $RhCl_3 \cdot 3H_2O$, $H_2PtCl_6 \cdot 6H_2O$ of "chemically pure" mark. Aqueous solutions of $H_2PtCl_6 \cdot 6H_2O$ and $RhCl_3 \cdot 3H_2O$ were applied by the adsorption method on the prepared support $\gamma$-$Al_2O_3$. The catalyst samples were filtered off and dried at 100–110 °C, to a constant weight. The reduction of supported catalysts was carried out in a quartz tube with electrical heating in a hydrogen stream at 200 °C for 4 h, then the catalysts were cooled in a hydrogen stream until they reached room temperature. The synthesized catalysts were non-pyrophoric, and were stored in the desiccator over calcium chloride.

#### 3.2.2. Catalytic Neutralizer

As the primary support, a heat-resistant X23U5 foil with a thickness of 50 μm was used, which was subjected to shirring and was rolled in the form of cylindrical blocks with a diameter of 45 mm and length of 90 mm. The final block support had 45 channels of 1 $cm^2$.

The prepared block metal support with the honeycomb channels structure was applied alongside a secondary support. The secondary support was a suspension containing aluminum salts (boehmite and aluminum nitrate). The impregnated slurry block supports were dried at temperatures up to 150 °C and then calcined at 500 °C for 2 h. The amount of secondary support was controlled by the weighting method and was about 20% of the block weight. If necessary, application of the secondary support was repeated.

Aqueous solutions of salts with compounds of the respective metals ($Co(NO_3)_2 \cdot 6H_2O$ or $H_2PtCl_6 \cdot 6H_2O$ + $Co(NO_3)_2 \cdot 6H_2O$) were impregnated by capacity on the prepared support. Afterward, the blocks were dried and calcined for 2 h at 500 °C. A solution containing noble metals ($H_2PtCl_6 \cdot 6H_2O$,) was prepared immediately before the impregnation by mixing a predetermined amount of the solution; for example, platinum chloride acid with distilled water. The noble metal content was 0.1% of the weight of the catalyst block. The content of Pt components was 0.5% and Co was 10% in terms of metal. Samples with additions of magnesium oxide, as a competitor to cobalt oxides, were prepared by the same procedure with a simultaneous co-impregnation from the total solution of salts of $Co(NO_3)_2 \cdot 6H_2O$ and $Mg(NO_3)_2 \cdot nH_2O$, with a content of 15% Co and 15% Mg.

### 3.3. Analyses and Instrumentation

The textural properties of the catalysts were studied using low-temperature nitrogen adsorption at −196 °C on the AccuSorb unit (Micrometrics, Norcross, GA, USA).

Analysis of the gasoline fractions before and after the reactor was conducted on a gas chromatograph Crystallux-4000M (LLC NPF "Meta-Chrom", Yoshkar-Ola, Russia), with a flame ionization detector.

$H_2$-TPR was carried out on the chromatograph "Chromatech-Crystal 5000", (JSC SDO Chromatec, Yoshkar-Ola, Mari El Republic, Russia). The detector was a katharometer. The TPR was carried out with a gas mixture of $H_2$-Ar with a hydrogen content of 5%. The flow rate was 30 $cm^3$/min and the heating rate was 5 °C/min. Heating was carried out in a linear mode up to 1200 °C, with subsequent thermostating at this temperature until the curve completely left the zero line.

TPD of hydrogen on the catalysts was carried out using equipment GDTD-24 AB firm of Setaram (France), to which a U-shaped furnace cell was adapted. TPD curves were recorded using the following technique: A weighed portion of the catalyst (0.25 g) was loaded into the cell and trained at room temperature in inert gas for 1 h (to remove physically adsorbed water). Then, a gas mixture of argon with hydrogen (7% $H_2$ in Ar), purified from impurities and water vapor, was passed through the sample. The linear temperature rise rate was 10 °C/min. The change of the hydrogen concentration in the support was determined on a "Cromodam" chromatograph. Some of the catalysts were preliminarily reduced in the flow of hydrogen at 473 K for 1 h, followed by exposure to air.

XRD was carried out on the diffractometer DRON-4-07 with Co-K$\alpha$ radiation (JSC Bourevestnik, Saint-Petersburg, Russia).

The EM of the samples was studied by two methods: (1) TEM on an "EM-125K" electron microscope (Sumy electron microscope plant, Sumy, Ukraine) by the coal replica method, with extraction using microdiffraction, and (2) SEM on a microscope JSM 6610LV (JEOL Ltd., Tokyo, Japan) with an accelerating voltage of 10 kV (magnification 25,000×).

### 3.4. Catalytic Experiments

The experiments for hydrodearomatization were carried out in a 0.5 L batch stirred reactor (Autoclave, from firm Amar Equipments Pvt. Ltd., Mumbai, India.) equipped with inlet and outlet valves for adding or removing gases and a valve for sampling at different reaction times. Hydrogenation experiments were carried out at a reaction temperature of 25–200 °C and 1.0–5.0 MPa. The reaction mixture was prepared by vigorously mixing 1 g of catalyst and 200 mL gasoline fraction. Before each reaction test, the catalyst was first reduced at 200 °C (linear heating rate 7 °C $min^{-1}$) for 2 h, using a pure hydrogen flow of 30 mL/min. The reduced catalyst was cooled to room temperature in a stream of hydrogen and immediately transferred to the reaction mixture. The reactor was sealed and purged with pure hydrogen three times. Then, the pressure was increased to the desired pressure, followed by an increase in temperature to the predetermined value.

To neutralize the exhaust gases of the ICE, the catalyst (cylindrical block with a diameter of 45 mm, length 90 mm) was placed in a metal frame, which was installed after the block for additional heating of gases coming from the exhaust collector of the ICE. In the process of neutralization, the conversion ($\alpha$) of the substance (CO and $CH_x$) was determined by the formula:

$$\alpha = C_{in} - C_{fin}/C_{in} \times 100\% \tag{1}$$

where $C_{in}$ and $C_{fin}$ are the initial and final concentrations of carbon monoxide or hydrocarbons in the sample volume.

A gasoline ICE KIPOR KG160 (Kipor, Wuxi Kama Power Co. plant, Wuxi, China, model: KG160) with the following characteristics was used: Engine power 3.3 kW, rotational speed 3600 rpm, engine volume 163 mL.

Exhaust gas sampling was carried out at 2500 rpm (according to the tachometer on the engine shaft). The measurements were carried out in flow-through installation, with determination of the gas composition with the gas analyzer "OPTIMA 7" (MRU GmbH, 74172 NSU–Obereisesheim, Neckarsulm, Germany) before and after the catalyst. The flow rate was 100,000 $h^{-1}$.

## 4. Conclusions

As long as vehicles use petroleum-based fuels, problems with air pollution due to toxic exhaust gas emissions will be an issue. The problem is worsening due to the deterioration of oil quality (heavy and high-sulfur oil), especially in Kazakhstan. The unsatisfactory quality of such fuel leads to incomplete combustion and a high toxic substances content in the vehicle exhaust gases. This environmental problem should be solved by creating environmentally-friendly vehicles, by improving the motor fuel quality and through efficient toxic exhaust treatment. Two gasoline fractions (after the reforming stage and without the additives) were taken from Kazakhstan Refinery plants (Atyrau and Pavlodar) and were subjected to the hydrodearomatization process over Rh-Pt(9:1)/$\gamma$-Al$_2$O$_3$ catalysts. The hydrotreated fuels were further tested in an ICE to determine the amount of harmful emissions. The final step of this work consisted of the neutralization of the formed toxic substances on metal block neutralizers. The research results showed that during the combustion process, benzene-free fuel with a reduced aromatic content gives 6.6–24.7% less toxic emissions compared to the initial gasoline fractions and achieves 100% CO removal and a decrease of hydrocarbons by 94% in the exhaust gases. Another positive outcome was the removal of olefins, which make the fuel unstable. Octane numbers practically do not decrease after catalytic treatment of gasoline fractions, due to the process of hydroisomerization of paraffinic hydrocarbons to isoalkanes. Effective catalytic systems were developed during the research processes: Bimetallic catalyst Rh-Pt(9:1)/$\gamma$-Al$_2$O$_3$ for hydrodearomatization and 10% Co + 0.5% Pt/Al$_2$O$_3$ supported on metal blocks with a honeycomb structure for neutralization. The catalysts with a low noble metal content (0.5%) showed high activity, with up to a 91–100% conversion rate. The study of catalysts by the EM, XRD, BET, TPD and TPR methods showed the factors responsible for the high activity of the catalysts. With further research, catalysts could be recommended for pilot tests at refineries in Kazakhstan and on motor vehicles.

**Author Contributions:** Conceptualization, A.M., A.S.; methodology, A.S. and A.M.; validation, N.K., M.K., A.B., A.U.; formal analysis, N.K., M.K.; investigation, K.R., M.K., A.U., A.B.; Writing—original draft, A.M., A.S.; Writing—Review & Editing, A.M., A.S.; Visualization, M.K., N.K.; Supervision, A.S., A.M. All authors have read and agreed to the published version of the manuscript.

**Funding:** This work was financially supported by the Ministry of Education and Science of the Republic of Kazakhstan (Project No. BR05236739).

**Acknowledgments:** We thank Karmyssov Kuralbek from "Atyrau Refinery" LLP and Ruslan Smagulov from "Pavlodar Oil Chemistry Refinery" LLP for supplying the gasoline fractions.

**Conflicts of Interest:** The authors declare no conflict of interest.

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
