# Peer review of "Catalytic Technologies for Solving Environmental Problems in the Production of Fuels and Motor Transport in Kazakhstan"

_catalysts, doi:10.3390/catal10101197_

Round 1

Reviewer 1 Report

The present manuscript demonstrates the hydrodearomatization of gasoline using Rh-Pt (9:1)/γ-Al2O3 catalysts prepared by the impregnation method. Almost 100% benzene removal and 50 % aromatics removal were achieved. While using 10% Co + 0.5% Pt/Al2O3 as a catalyst, complete oxidation of CO to CO2 in the exhaust was observed. The catalysts were characterized using various characterization tools such as SEM, TEM, TPR, and XRD. A detailed neutralization and TPR study are performed. Overall, the manuscript is well written. However, a few parts have general information and do not need a detailed explanation. The manuscript can be re-edit to make it focused and concise. After that, the manuscript can be considered for publication. Some minor comments are:

  1. Reduce the basic information from the introduction.
  2. The abstract should be rewritten, and abbreviations should be explained at first use.
  3. The manuscript title is too long
  4. Figure caption a and b should be in Figure.
  5. SEM scale bar is missing.
  6. In the table instead of a comma, the point should be used for numbers.
  7. A flow chart showing different systems and neutralization and conversion using each catalyst can be included for better clarity.

Author Response

Dear Reviewer,

Please see the attachment below

Reviewer 2 Report

This manuscript reported the combines processes for improvement of the quality of 2 gasoline fractions from Kazakhstan refineries via hydrodearomatization over Rh-Pt(9:1)/gAl2O3  and  exhaust gases cleaning by gases neutralizing on 10 % Co + 0.5 % Pt/Al2O3 catalyst. They already had two papers with similar topic, which are

  1. Massenova, A.T.; Kalykberdiyev, M.K.; Sass, A.S.; Kenzin, N.R.; Kanatbayev, E.; Baiken A. Catalytic Technologies for the production of Eco-friendly Gasolines and Reducing the Toxicity of Vehicle Exhaust Gaese. Orient J Chem 2019; 35 (1).
  2. Massenova, A.T.; Kalykberdiyev, M.K.; Sass, A.S.; Kenzin, N.R.; Kanatbayev, E.T.; Tsygankov, V.P. Hydrogenation of aromatic hydrocarbons in gasoline fractions over supported catalysts under pressure. News of the
    national Academy of sciences of the Republic of Kazakhstan, Series chemistry and technology
    2018, 5, 146-153.

However, compared to their previous papers, this paper barely contains any new knowledge. Therefore, I cannot recommend the publication of this paper in Catalysts.

Author Response

Dear Reviewer,

please see attachment below

Reviewer 3 Report

The paper analyses two fractions from Refineries of Kazakhstan in order to improve their quality by  hydrodearomatization catalysts Rh-Pd/g-alumina with different compositions, and the hydro-treated fuels were tested in an internal combustion engine to analyze the produced toxic compounds and compare them with initial fuels. CO and Hydrocarbons on the combustion engine exaust gas were abated on an oxidation catalyst Co-Pt/alumina.

The paper gives an extensive work on the chemical analysis of two specific refinery fractions, testing them in a real motor engine, and tests catalysts with different composition. The characterization of catalysts is appropriate, but the EM analysis is limited to one sample, not allowing the comparison the metal dispersion for different metal concentrations.

The criticisms of this paper are related to:

1) XRD discussion and images have to be improved

2) the conclusion paragraph have to be modified and deepen.

3) EM analysis should be extended

4) The text needs a revision about the English language, particularly about syntax.

1) Line 138-141: the purpose of the paper should be better explained, with more details in the introduction. The required details have been reported in the conclusion instead of the introduction. For clarity, I suggest to move them in this paragraph.

2) the conclusion paragraph needs a deep revision. It reports information already reported in the introduction and reports the purpose of the research that for clarity have to be removed from this paragraph (lines 514-521) + ( lines 528-530) + (lines 535-537).

In addition, it does not extensively report the conclusions about of the addition of  Pt and Mg, that was one of the purpose of the research (line 545 cites the study without the results).

3) In the text authors have to substitute the numbers with the word: i.e. 2 fractions, 2 sets, 4 on the background…. Etc..

4) “Stable catalisate” this frequently reported expression is not commonly used in catalysis research reports and it is not clear what it means: add a correct clear definition of that. why the capital letter?

5) XDR analysis: Line 249: diffraction lines are reported as d in Angstrom from ASTM card.   There is not correspondence with the figure 5 that reports the peaks at the 2Theta angle. It is strictly necessary substitute them with the corresponding 2Theta position from JCPDS reference card. In the figure 5 it is best to add a symbol on the lines of alumina or the hlm plane as done in the figures commonly reported in the literature, to favor the comparison by other authors.

6) Paragraph 3.2.1: diffraction lines are reported without units. In addition, they do not correspond to the figure 6-7. 5 reporting 2theta positions. Authors have to refer to 2theta values of the figures. In the figures they have to report symbols or hlm.

7) Fig 6 and 7: diffraction lines are attributed to the cobalt, cobalt alluminate and alumina species, and for clarity of the reader, they should be distinguished with a different symbols for the different species. Fig 6, 7, 8: there is not the unit in the x axes

8) lines 231-232: authors talk about “homogeneity of Hydrogen”. In a catalyst characterization, the attention should be on the surface, so on the homogeneity of the sites instead of the adsorbed molecule. I suggest to add some considerations about the homogeneity of the adsorption sites.

9) Lines 501-504 and paragraph 2.3: why you define the reaction of oxidation as NEUTRALIZATION process?. Are authors sure that it is appropriate? I suggest to reconsider this definition.

10) EM analysis: the images are referred to which Rh-Pt (9:1) sample? EM analysis should be extended to the other metal concentrations.

11) Paper needs a deep revision of the English language:

These are some English sentences to revise and correct in the grammar :

Line 15: Study following

Lines 17-19

Lines 23-24

Lines 27-28

Lines 35-36

Lines 41: is determined

Line 93: an increase not only the activity

Lines 111-113

Line 337: At co-deposition

Line 338, and 339 and check others…: reflexes instead of reflections

Author Response

(The authors gave the same response as above.)

Round 2

Reviewer 2 Report

After revising, the quality of the manuscript is improved, except some minor spelling and text errors. Therefore, the revised manuscript can be published after text editing.

Author Response

We have done English editing of the article. And we are sending you this revised article.

Reviewer 3 Report

I report the authors answer to my suggestion in point 5:

Referee point 5: XRD analysis: Line 249: diffraction lines are reported as d in Angstrom from ASTM card. There is not correspondence with the figure 5 that reports the peaks at the 2Theta angle. It is strictly necessary substitute them with the corresponding 2Theta position from JCPDS reference card. In the figure 5 it is best to add a symbol on the lines of alumina or the hlm plane as done in the figures commonly reported in the literature, to favor the comparison by other authors.

 Authors' Answer: ASTM replaced with JCPDS throughout the text. All lines where ASTM were replaced to JCPDS: Line 232, line 233, line 294, line 298, line 299, line 312, line 317, line 318, line 330, line 331, line 332, line 338. (All these lines for switched off “Track changes”) In all the XRD patterns, we did not apply bar diagrams, because there are simultaneously several spinel phases, which differ little from each other in the position of the reflections. Therefore, the figure is difficult to read due to the abundance of phases of the same parameters. Since in more complex figures with a large number of identical phases, we did not make bar diagrams, in a simpler figure (Figure 5) we also did not apply these bar diagrams to preserve the uniformity of the presentation of the material, but limited to describing their position in the text with reference to the corresponding JCPDS.

Referee replay to the authors answer:

The authors have chosen to ignore my suggestion 5 or they have misunderstood the meaning of my comment. The substitution should not simply be for the acronym JCPDS instead of ASTM, but the suggestion was to change the Angstrom positions to the 2theta position, to match the figure, or to give more information, to change the corresponding plane to angstrom hlm. This is how an XRD diffrattogram is generally presented in scientific papers. I did not talk at all about bar diagrams, that would be totally useless.

Unfortunately, the authors did not agree to improve their work about this issue.

The other comments were sufficiently accepted, improving the paper in those aspects.

Author Response

Reply to your 5 comment

 We have two X-ray apparatus, working on tubes with Cu and Co anodes. The Angles of the reflection position change depending on the anode material. The Interplanar distances remain constant for any anodes. Therefore, it is more convenient to indicate the position of reflections not in angles, but in interplanar distances expressed in Å. In catalogs, the positions of reflections are also indicated in Interplanar distances in Å.